# Emerging Strategies for Enhancing Propionate Conversion in Anaerobic Digestion: A Review

**DOI:** 10.3390/molecules28093883

**Published:** 2023-05-04

**Authors:** Lan Mu, Yifan Wang, Fenglian Xu, Jinhe Li, Junyu Tao, Yunan Sun, Yingjin Song, Zhaodan Duan, Siyi Li, Guanyi Chen

**Affiliations:** 1School of Mechanical Engineering, Tianjin University of Commerce, Tianjin 300134, China; mulan@tjcu.edu.cn (L.M.);; 2School of Biotechnology and Food Science, Tianjin University of Commerce, Tianjin 300134, China; 3Tianjin Capital Environmental Protection Group Co., Ltd., Tianjin 300133, China; 4School of Environmental Science and Engineering, Tianjin University, Tianjin 300072, China; yingjin@tju.edu.cn

**Keywords:** anaerobic digestion, methane, biogas, propionate degradation, VFAs, trace elements, carbon material, DIET

## Abstract

Anaerobic digestion (AD) is a triple-benefit biotechnology for organic waste treatment, renewable production, and carbon emission reduction. In the process of anaerobic digestion, pH, temperature, organic load, ammonia nitrogen, VFAs, and other factors affect fermentation efficiency and stability. The balance between the generation and consumption of volatile fatty acids (VFAs) in the anaerobic digestion process is the key to stable AD operation. However, the accumulation of VFAs frequently occurs, especially propionate, because its oxidation has the highest Gibbs free energy when compared to other VFAs. In order to solve this problem, some strategies, including buffering addition, suspension of feeding, decreased organic loading rate, and so on, have been proposed. Emerging methods, such as bioaugmentation, supplementary trace elements, the addition of electronic receptors, conductive materials, and the degasification of dissolved hydrogen, have been recently researched, presenting promising results. But the efficacy of these methods still requires further studies and tests regarding full-scale application. The main objective of this paper is to provide a comprehensive review of the mechanisms of propionate generation, the metabolic pathways and the influencing factors during the AD process, and the recent literature regarding the experimental research related to the efficacy of various strategies for enhancing propionate biodegradation. In addition, the issues that must be addressed in the future and the focus of future research are identified, and the potential directions for future development are predicted.

## 1. Introduction

Finite fossil fuel reserves and constantly rising fossil fuel prices, as well as contaminated water, air, and land, have all encouraged a severe energy crisis and environmental challenge related to the burning of fossil fuels [1,2]. Mounting concerns about global warming, which resulted from massive CO_2_ emissions, have also driven researchers to seek alternative, sustainable energy sources (Figure 1). In this context, renewable energy has continued to gain momentum within the past few decades, from 1.0% (share of global primary energy consumption) in 1972 to 2.0% in 2007 and 6.2% in 2021 (Figure 1). The average increasing rate of renewable energy sources based on global energy consumption is 9.24%, and the share of renewable energy is predicted to reach 30% in 2035, for which abundant biomass and organic waste are important resources.

Anaerobic digestion is widely used in the treatment of organic waste and organic wastewater [3,4,5]. According to the concentration of total solid (TS) feed content, anaerobic fermentation can be divided into wet anaerobic and dry anaerobic processes [6,7]; according to the use of fermentation tanks, this can be divided into continuously stirred tank reactors (CSTRs) [8], upflow anaerobic sludge beds (UASBs) [9], internal circulation (IC) [10], expanded granular sludge beds (EGSBs) [11], upflow solid reactors (USRs) [12], etc.; according to the temperature of anaerobic fermentation, the process can be divided into thermophilic, mesophilic, and psychrophilic digestion. It is a relatively energy-saving and efficient treatment method. Carbon emissions from composting, landfill, and waste incineration range from 61 to 1010 kg CO_2_-eq/t FW, which is much higher than AD. In addition, the biogas generated (50–75% CH_4_ and 25–45% CO_2_) [13] can be used for heat production, power generation, or the purification of natural gas; the waste residue generated by its fermentation contains a large amount of trace metal elements and nutrients that can stimulate the growth and development of plants, so it can also be used as an organic fertilizer [14]; the derived biochar can be recycled to regulate the stability of the anaerobic digestive system, which are known as biochar-amended digestors [14].

In recent years, more and more countries and regions have begun to widely use this technology for the production of bioenergy. For instance, Europe counted over 18,943 commercial combined heat and power (CHP) plants in 2019. In total, they produced 167 TWh from biogas for CHP utilization and 26 TWh from purified methane for injection into the municipal grid. Germany has more than 8000 AD plants, which generate approximately 4.0 × 10^10^ kWh/year [15]; Australia intends to achieve 5.6 × 10^10^ kWh/year bioenergy generation through AD plants in 2050 [16]. In the USA, the government provides AD plants with a 1.1 ￠/kWh tax credit for the first 10 years to support the development of AD technology [17]. The anaerobic digestion treatment of organic waste has prevailed in recent years. Cambi^®^ thermal hydrolysis + anaerobic digestion has developed fast and has quickly taken over the market in the past 10 years. The installed AD plants include the Blue Plains wastewater treatment plants in the USA, the Chertsey, Riverside, Crawley, Beckton, Crossness, Long Reach, and Oxford wastewater treatment plants in the UK, the Xiaohongmen, Gaobeidian, Gao’antun plants in China, and so on. In addition, the co-digestion of multiple substrates, which can compensate for the disadvantages of single digestion, was adopted in many cases [18,19]. Although the AD plants installed in developing countries (China, India, Brazil) are mainly used for the treatment of municipal sewage sludge, the practice is expected to grow rapidly in the next few decades [20].

The anaerobic digestion process involves multistep complex biochemical reactions, which can be mainly divided into three stages (Figure 2): (i) the hydrolysis of complex polymers, (ii) the fermentation of hydrolytic products to short VFAs, formate, CO_2_, and H_2_, and (iii) the conversion of fermented products into CH_4_ by methanogenic bacteria. Since the substrates that can be utilized by methanogenic bacteria are limited within acetate, methyl group-containing compounds, such as CO_2_ and H_2_ [21], and other VFAs, like propionate, butyrate, valerate, or ethanol, have to be converted to acetate and H_2_ before being taken up by methanogenic bacteria. This process is known as acetogenesis and plays a vital role because it is responsible for 76% of the transformation of the reduced organics [4]. Among the VFAs, propionate degradation is critical due to the fact that about 6–35% of the total methanogenesis is via propionate conversion and the oxidation of propionate to acetate, and H_2_ is often considered as the limiting step. This is because the oxidation of propionate has the highest Gibbs free energy (+76 kJ/mol) compared to other VFAs. The oxidation of propionate is more energetically unfavorable unless the produced acetate and H_2_ are synchronously consumed by acetate- and H_2_-utilizing bacteria [22]. However, in reality, the balance between propionate oxidization and utilization is often unco-ordinated when encountered with a putrefactive substrate, high OLR, low C/N ratio, unbefitting inoculum, and other issues. Propionate easily accumulates in these anaerobic digesters; thus, it induces the inhibition of anaerobic micro-organisms and the deterioration of digesters, which has been frequently reported by many researchers [23,24,25,26].

Until now, the anaerobic digestion of biomass or organic waste has been extensively reviewed in different substrates [3,27,28,29,30,31], reactor configurations [32], pretreatment strategies [33], co-digestion performance [34], models [35], inhibition factors or toxicants [36], promotion measures, like trace metals [4] or bioaugmentation [37], microbial characteristic [38,39], biogas purification [40], and so on. Although some papers included the inhibition effect or the degradation performance of propionate, few articles reviewed the development of engineering-enhancing strategies for propionate degradation in anaerobic digestion. Given this and the critical role of propionate in anaerobic digestion, comprehensive information on its generation and metabolism mechanism have been outlined in this article, and the parameters influencing the conversion of propionate were discussed, together with engineering strategies, with the aim of promoting the propionate biodegradation proposed by different researchers. Some perspectives pertaining to propionate degradation were also provided.

## 2. Production and Metabolism of Propionate in Anaerobic Digesters

### 2.1. Production of Propionate

Before the biomass is fermented to acids, the complicated components have to be hydrolyzed (Figure 2). Hydrolysis is a series of biochemical reactions that decompose biomass polymers (such as polysaccharides, proteins, and lipids) into monomers or oligomers. These reactions are catalyzed by a series of distinct functional extracellular enzymes, including amylase, protease, lipase, cellulase, hemi-cellulase, and xylanases. The involved micro-organism can be mainly classified into two phyla: *Bacteroidetes* and *Firmicutes* [36]. Then hydrolyzed products are converted to VFAs, formate, H_2_, and CO_2_ by a guild of fermentative bacteria. These fermentative organisms belong to *Chloroflexi*, *Bacteroidetes*, *Proteobacteria*, *Firmicutes*, *Synergist*, and *Actinobacteria* [41]. Among them, the species *Geobacter*, *Pelobacter*, *Streptomyces*, *Sorangium*, *Desulfatibacillim*, *Rhodopseudomonas*, *Mycobacterium*, *Acidovorax*, *Pseudomonas*, and *Ralstonia* were reported to be engaged in propionate formation in anaerobic digesters [42].

Pyruvate plays a pivotal role in the network of various metabolic pathways of polysaccharides, proteins, and lipids. In fermentative bacteria, there are two main pathways to propionate formation from pyruvate, as shown in Figure 3. The first one is the acrylate pathway with lactate as an intermediate. The pyruvate is reduced by lactate dehydrogenase (LDHA) to lactate, which is then activated to lactoyl-CoA by propionyl-CoA transferase (PCT). Lactoyl-CoA is dehydrated and generates acryloyl-CoA. Then, acryloyl-CoA is further reduced by acrylyl-CoA reductase (ACR) to propionyl-CoA, which is the active form of propionate. This pathway can be carried out by some amino-acid utilizing *Clostridia*, such as *Clostridium propionicum* [43]. The other pathway is the methylmalonyl-CoA pathway, in which pyruvate is converted to oxaloacetate by pyruvate carboxylase (PC). Then, oxaloacetate undergoes a series of enzymatic catalytic reactions and generates propionyl-CoA. This process could be performed by many acidogenic bacteria, like *Corynebacteria*, *Propionibacterium*, and *Bifidobacterium* [44].

### 2.2. Metabolism of Propionate

Propionate cannot be directly used by methanogenic bacteria, but it has to be converted to acetate and H_2_ by acetogens at first. The metabolism route of propionate is as stated in the equations (i), (M), and (i + M) in Table 1. The Gibbs free energy of propionate oxidation is +76.1 kJ/mol, which is thermodynamically unfavorable [45]. Only when coupled with the oxidation reaction of H_2_ does the overall reaction of propionate become exothermic (equation i + M), which means the H_2_ generated should be consumed synchronously and be maintained at a very low partial pressure (lower than 10^−4^ atm). This co-operative relationship between propionate-oxidizing bacteria and H_2_-consuming bacteria is called syntrophic association.

The first syntrophic propionate-oxidizing bacteria were reported by Boone [46] based on the co-culture experiment of syntrophobacter wolinii with the H_2_-consuming bacteria *Desulfovibrio* sp. Since then, some other species have been identified, which mainly are assigned to *Syntrophobacter*, *Pelotomaculum*, and *Smithella propionica*, with two pathways of propionate metabolism, as shown in Figure 4. One is the methyl-malonyl-CoA pathway (MMC), which is similar to the reversion process of propionate formation and prevails in most propionate-oxidizing bacteria. In the MMC process, propionate is converted to the intermediates succinate, fumarate, malate, oxaloacetate, and pyruvate, which are further catalyzed by malate degydrogenase (MDH) and phosphotransacetylase-acetate kinase (PTA-ACK) into acetate. The Gibbs free energy of the oxidation of succinate to fumarate is +55.96 kJ/mol, which is thermodynamically unfavorable and is considered the limiting step. The other route of propionate decomposition is the dismutation pathway (Di-pathway), which was found in a syntroph Smithella propionica by Liu et al. [47] and was verified using ^13^C-NMR spectroscopy by de Bok et al. [48]. This route includes the condensation of the C_2_ of propionate to the carboxyl of another propionate molecule or its derivative, the rearrangement of the methyl group, the transfer of the oxygen to the C_3_ of the intermediate, and the cleavage of 3-ketohexanoate, yielding butyrate and acetate [48].

Direct interspecific electron transfer, with the aid of electrically conductive pili and the associated c-type cytochrome or artificial conductive materials [49,50,51], was reported in recent years. In this pattern, very little hydrogen (traditional electron shipper) was generated, and the electron of the VFAs was directly transferred to aceticlastic methanogens and oxidized to methane and carbon dioxide. However, few pieces of evidence have been reported on the direct interspecific electron transfer of propionate; these are reviewed in Section 4.

## 3. Critical Parameters Influencing the Biodegradation of Propionate

Temperature: Anaerobic digestion can be subdivided into three categories based on different operational temperatures and particular microbial communities: psychrophilic (10–20 °C), mesophilic (30–40 °C), and thermophilic (55–70 °C) [32,52]. In general, psychrophilic anaerobic digestion is rarely used in applications, and propionate metabolism has been reported to be significantly inhibited under psychrophilic conditions [53]. From 20–35 °C, an elevated temperature was beneficial to propionate conversion [53], indicating the optimum temperature of POBs involved. It is well acknowledged that thermophilic anaerobic digestion often has a higher organic hydrolysis rate and conversion rate, but the inhibition of propionate conversion in thermophilic digesters was reported in some studies. For instance, Jiang et al. [54] found that compared with mesophilic digesters, high levels of propionate were observed in thermophilic digesters, which was attributed to low affinities between propionate and the propionate-oxidizing bacteria of thermophilic species. At an organic load rate (OLR) of 10.0 kg·COD/(m^3^·d), the methane production is 310 mL/g·COD. A similar result was observed by Yang et al. [55]. Li et al. [56] also observed the strong inhibition of propionate oxidation in thermophilic digesters with a lag phase of longer than 17 days. The maximum rates of methane production (R_max_) fitted using the Gompertz model were 27–62% lower than those using mesophilic digesters. In contrast, Zhao et al. [57] investigated the effect of propionic acid on the activity of methanogenesis. When the concentration of propionic acid is higher than 5000 mg/L, the AD system is completely inhibited. When compared with mesophilic digesters, the propionic acid degradation rate of the thermophilic digestor is relatively higher, with a higher R_max_ fitted using the Gompertz model and higher hydrolysis rates fitted using a first-order dynamic model.

pH: pH plays a direct role in microbial growth and metabolism, as well as in propionate-oxidizing bacteria, since changing the pH causes a change in electric charge on the cell membrane, which thereafter influences the assimilation of nutrients and the activities of enzymes [58]. Li et al. [59] found that the propionate degradation rate was much higher at pH 7.0–8.5 than at pH 6.5 or below. Similarly, Zhang et al. [60] used 2000 mg/L propionic acid as the only carbon source in a UASB reactor. When the pH value was controlled at 6.8–7.5, the high propionate removal of the sludge blanket (81.5–90%) was observed, and the methane content and biogas productivity were maintained at 55.2–69.3% and 22.8–26.7 L/day; when pH was controlled at 6.0, 5.5, and 5.0, only 49.1%, 33.8%, and 16.7% of propionate was degraded, the methane content decreased by 5.1% and 33.7%, respectively, and the biogas productivity decreased to 15.3 L/day. When the pH value is below 4.5, propionate degradation almost does not occur, and biogas production almost stops. The decline in the oxidation performance of propionate was related to the significant reduction in propionate oxidation bacteria and acetoclastic methanogens, which were more sensitive to low pH [60].

Oxidation-reduction potential (ORP): ORP can significantly influence the fermentation type in the acidogenic phase in anaerobic digestion. It is a key factor affecting the production and consumption of VFAs. By altering ORP, the ratio of NAD/NADH is changed, cell metabolism is affected, as well as the distribution of fermentation end products is also altered [61]. The optimal ORP range for acid-producing fermentation is between −100 mV and −300 mV. The formation of propionate occurs under ORP conditions above −278 mV [62]. In an anaerobic reactor with iron added, the negative value of ORP is greater. If ORP is reduced to −300 mV, it will inhibit the formation of propionate and increase the production of CH_4_. Therefore, ensuring the reduction in atmosphere is critical to preventing the propionate fermentation route. Reducing additives, such as zero-valent iron, can be added to anaerobic digesters to prevent the formation of propionate and accelerate its degradation [63].

Acetate concentration and H_2_ pressure: Propionate cannot be directly taken up by methanogens but can be decomposed to acetate, H_2_, and CO_2_ at first by acetogenic bacteria. On the one hand, thermodynamically speaking, the conversion of propionate to acetate and H_2_ is only possible when the H_2_ partial pressure is below 10^−4^ atm, and the acetate concentration is lower than 23 mM [64]. A high hydrogen partial pressure will also block the acetate oxidation. Many results manifested the severe inhibition of the digesters from high H_2_ partial pressures [65,66]. In Cazier’s research, the influence of H_2_ and CO_2_ pressure separation on solid-state AD was explored [66]. When the H_2_ partial pressure was increased from 0 to 600 mbar, the CH_4_ yield increased from 23 ± 4 mL CH_4_/g TS to 28 ± 6 mL CH_4_/g TS. When further increasing the H_2_ partial pressure to 1555 mbars, the methane yield of CH_4_ decreased from 28 ± 6 mL of CH_4_/g TS to 9 ± 1 mL of CH_4_/g TS. More specifically, when H_2_ is higher than 800–900 mbar, methane production decreases very fast. On the other hand, H_2_ cleavage is only possible when the H_2_ partial pressure is above 10^−6^ atm [67]. Thus, in order to ensure the metabolism of propionate, the H_2_ partial pressure should be maintained between the narrow range of 10^−6^–10^−4^ atm.

Propionate concentration: The organic loading rate (OLR) is a crucial operation parameter for an anaerobic digester. Propionate accumulation at a high OLR or short hydrolytic retention time (HRT) was frequently reported in the digestion of many substrates, such as food waste [63,68]. This could be attributed to the imbalance between propionate production and conversion rate, but the conversion rate of propionate at different propionate OLRs was rarely reported. When using digestate from a semi-continuous anaerobic digester with coffee powder as substrate, Zhao et al. [57] found that the hydrolysis content, k, modeled using a first-order dynamic model decreased from 0.82 to 0.13 when propionate loading increased from 0.5 to 8.0 g COD/L in mesophilic conditions, and a decrease from 0.89 to 0.31 in thermophilic conditions, indicating that elevated propionate concentration in digesters inhibits the activity of POBs. Rafika et al. [69] researched the performance of a semi-continuous anaerobic digestor under different OLRs. When the OLR increased from 3.44 g VS/L·d to 14.6 g VS/L·d, the methane productivity presented earlier increased and later decreased. Under an OLR of 4.25 gVS/L.d, one digestor showed a high energy potential of 530 L CH_4_/kg VS, while further increasing the OLR lead to propionate accumulation.

Digester configuration: When considering the hydrogen transfer mechanism and the low solubility of hydrogen in a liquid, the short distances between hydrogen-producing bacteria and the consuming species would be favorable. The estimation value for H_2_ transfer distances in a digester was about 11 μm, which is equal to about 10 bacterial widths [70]. Instead of dispersed sludge via granule sludge or the immobilization of anaerobic sludge, this exactly meets the aim, with the syntrophic species being close together. Upflow anaerobic sludge blanket (UASB) reactors, expanded granular sludge bed (EGSB) reactors, and their derivatives are the most widely installed systems using granular sludge in the treatment of municipal wastewater [32]. A sludge granule is a multi-layered aggregate with acetoclastic methanogens in the interior and hydrogen/formate-producing acetogens and syntrophic hydrogen/formate-consuming methanogens on the surface [24].

An appropriate kinetic model can analyze the changes in the process of an anaerobic digestion reaction, which is conducive to the accurate grasp of the reaction results, improving the stability of the system and providing guidance for the operation and control of anaerobic digestion. At present, dynamic models have been studied and applied as follows: a first-order kinetic model [71], a modified Gompertz model [72], anaerobic digestion model No. 1 (ADM1) [73], a two-phase exponential model [74], and a multi-stag [74] model. As a conventional model, the first-order dynamic model is used to depict the biogas yield process of different biomass wastes [75,76]. The formation of biogas is related to the activity of methanogens during anaerobic digestion; thus, the modified Gompertz model, which is derived from the growth model of mixed populations, was considered a good non-linear regression model [77]. The fitting error of the theoretical and practical results of the first-order model is larger than that of the theoretical and experimental results of the modified Gompertz model [78]. The modified Gompertz model can be used to obtain the retention time of biogas generation and the maximum methane yield. ADM1, another commonly used kinetic model, makes the fitting precision between the experimental results and the simulated values smaller, which gives simulated concentrations of the soluble organic components [79].

Existing studies show that the superposition model for the co-digestion of different biomasses can more accurately predict the potential of methane production. Wang et al. [80] used the superposition model to obtain a good fit for the co-digestion of pig manure and kitchen waste (R^2^ = 0.99). Adarme et al. [81] proposed the two-phase exponential model, which can also be used to describe methane production. In recent years, a machine learning model was introduced to accurately predict the kinetic parameters in anaerobic digestion models. In a previous study, Ge et al. [82] found that after model optimization, the average R^2^ for predicting seven kinetic parameters, including disintegration constant (K_dis_), hydrolysis constant of carbohydrates (K_hyd-CH_), the half-saturation constants of acetate (K_s_ac_), the half-saturation constants of monosaccharide (K_s_su_), etc., reached 0.92, and the root mean square error reached 0.167.

## 4. Engineering Strategies for Enhancing the Biodegradation of Propionate

### 4.1. Buffering Addition

Buffering addition is a direct way to increase the buffering capacity of an anaerobic digestion system and maintain a moderate pH value, which is important for propionate-degrading bacteria. Bicarbonate or phosphate is the most commonly used buffering material. In kitchen waste, it is easy to accumulate propionate via high organic loading digestion due to its corruptibility. In an anaerobic digestor containing solid residual kitchen waste, the addition of 1000 mg/L NaHCO_3_ increased the conversion rate of propionate by 50% under high organic loading but low inoculum ratio (I/S = 1:3.5), and the anaerobic digestion capacity without acidification increased up to 33.3% [83]. The effectiveness of bicarbonate in enhancing propionate degradation might be considered due to the alternation of dominant methanogens and Gibbs free energy [84]. The addition of bicarbonate can favor hydrogenotrophic methanogenesis, which thermodynamically benefits the degradation of propionate [85]. Zhang et al. [86] directly proved that the Gibbs free energy of propionate degradation decreased from 0.12 kJ/mol (without bicarbonate addition) to −15.29 kJ/mol with 0.05 mol bicarbonate/L and to −18.73 kJ/mol with 0.20 mol bicarbonate/L addition, which made syntrophic propionate degradation more feasible. In accordance with the higher propionate degradation rates, *synytophobacter sulfatireducens*, a propionate-oxidizing bacterium, was enriched by moderate bicarbonate addition [86]. Zhang et al. [87] also observed that with alkali (lime mud) addition, the amount of *methanobrevibacter,* which was reported as an acid-tolerate and hydrogenotrophic methanogen, was enriched and became the dominant archeae. With the addition of lime mud, increased from 2.0 g/L to 10.0 g/L, the carbon conversion rate (carbon in the biogas generated from unit feedstock/carbon in the unit feedstock) increased by 64.3% (1.4% vs. 2.3%).

Although the high concentration of bicarbonate addition can serve as an alternative in a VFAs/propionate overloading digester, excessive Na^+^ can be adverse to propionate-utilizing micro-organisms due to toxicity [88]. Although bicarbonate supplementation favored the hydrogenotrophic methanogens, excessive bicarbonate supplementation also caused an increase in Gibbs free energy in the methanogenesis of formate and acetate, which inhibited propionate degradation in reverse [86]. The boundary concentration of Na^+^ is about 3.5–8 g/L, as reviewed by Lin et al. [85]. In addition, in some cases of continuous operation, alkali addition cannot fundamentally solve the problem of propionate accumulation but only delays the failure of the process, with almost no improvement in the carbon conversion rate [63]. These results indicated that alkali addition was just a temporary strategy or was more effective in batch conditions. In addition, the supplementation of NaHCO_3_ or NaOH at the front end may lead to high Na solid residue and ash content, which is difficult to deal with.

### 4.2. Bioaugmentation

In principle, the conversion and degradation of a certain substrate in an anaerobic digestion system is related to a very specific metabolic process implemented by a mixture of specialized micro-organisms. Thus, the extra addition of specialized micro-organisms with the desired functions is a feasible approach to improve performance, which was defined as bioaugmentation. The concept of bioaugmentation started much earlier, but it has begun to receive attention in recent years for relieving the start-up of anaerobic digestion suffering from high acetic acids loads [89], preventing or shortening the recovery time of anaerobic digesters stressed by overloading or toxicants [90,91], improving system stability [92], enhancing hydrolysis and methane production from lipids waste [93], food waste [94], ammonia-rich substrate [95,96,97], lignocellulosic residues [98,99], municipal sludge [100], and so on.

The introduction of propionate-degradation or hydrogen-utilizing micro-organisms into an anaerobic digestion system can accelerate the degradation metabolism of propionate. Tale et al. [101] obtained a rapid propionate-utilizing enrichment culture over 580 days of semi-continuous acclimation operation, fed with propionate. When it was added into transiently overloaded digesters, the results showed a stronger performance for the degradation of acetate, propionate, and butyrate in this bioaugmented overloaded digester, and the recovery time was shortened by 25 days compared to a non-bioaugmented overloaded digester. By monitoring the difference in sCOD removal and the methane generation rate, it was found that the influence of bioaugmentation could persist for more than 12 SRTs after organic shock overload. The presence of hydrogenotrophic methanogens closely related to *Methanospirillum hungatei* and *Methanobacterium beijingense* was thought to be associated with a high VFAs degradation rate and methane productivity. In another study by Tale et al. [102], it was demonstrated that a bioaugmentation trial using methanogenic propionate-enrichment cultures obtained using limited aeration (25 mg O_2_/L day) had a higher abundance of *Methanospirillum hungatei*. Comparatively, *Methanolinea tarda* dominates the organisms in common acclimation or no-bioaugmentation digesters. Due to *Methanospirillum hungatei* having the highest growth rate and substrate utilization rate among hydrogenotrophic methanogens, a bioaugmented digester showed more rapid hydrogen consuming and more complete propionate degradation when using it, and thus higher COD removal and faster recovery after shock overload. By inoculating a culture using a mixture of sludge and cow dung and using sole propionate as a feed source, Acharya et al. [92] obtained a propionate-degradation enriched culture and inoculated it in a mesophilic two-stage reactor for the treatment of simulation wastewater (Figure 2). It was found that *Methanosarcinaceae* dominated the enriched culture and the methanogenic stage in the digester when added. The specific methane activity (SMA) value of the bioaugmented digester was seven times higher than that of the control digester, with the hydrogentrophic methanogenic activity being four times higher. As a result, the hydrogen partial pressure in the bioaugmented digester remained lower when compared to the control digester, as well as the propionate concentration, while acetic acid dominated the VFAs, which is preferred for methanogens. Thus, a lower effluent sCOD content and a higher CH_4_ content were observed in these bioaugmented digesters.

Ma et al. [103] designed and tested a separate EPAD (enhanced propionate acid degradation, Figure 5) system to help with recovery in high propionate accumulation CSTRs (continuous stirred tank reactors), based on the concept that the consortium acclimated to propionate degradation would have a higher degrading rate. The results showed the stop feeding of the control CSTR had no effect on decreasing the residual propionate concentration, and propionate concentration remained high even after 30 or 70 days of self-recovery, while the CSTR connected to EPAD successfully recovered due to the accelerated degradation of propionate. The evaluation results of up-scaling the EPAD system suggested that EPAD, with a volume of 2% in the full-scale digester, would be feasible as a mobile unit. This concept might provide an option for the practical application of bioaugmentation.

Although bioaugmentation is a promising approach to enhance propionate degradation in anaerobic digestion and improve the performance of a digester, especially when preventing or recovering from shock overload in a digester, there is still an argument about whether there are differences between bioaugmentation using extra micro-organisms and acclimation, which can be established by micro-organisms in the system itself [104]. In addition, extending this to the existing pilot or large-scale digesters for application should also be addressed in the future.

### 4.3. Supplementary Trace Elements

Supplementary trace elements represent an effective approach to preventing anaerobic digestion from VFAs accumulation and maintaining system stability [76,105,106,107]. The positive effect of trace elements in improving propionate degradation is mainly due to the acceleration of the growth rate of syntrophic hydrogenotrophic methanogens. Fe, Ni, and Co play key roles in propionate degradation. The supplementation of Fe + Co + Ni in a cow manure anaerobic digester led to a significant increase in the carbon conversion rate (from 1.3% to 2.3%) [76]. Fe is required for pyruvate-ferredoxin oxidoreductase, which catalyzes the oxidative decarboxylation of pyruvate and contains Fe–S clusters [108], and is also an essential element for formyl-MF-dehyogenase, which is an important enzyme in hydrogen-type methanognesis. Co is needed in vitamin B12, which is reported to bind to co-enzyme M [CoM] methylase that catalyzes a methy-transferring reaction in both aceticlastic and hydrogenotrophic methanogens [109]; Mo is required for formate dehydrogenase (FDH, Figure 2) and is involved in syntrophic propionate metabolism [38]; Ni is incorporated into co-enzyme F_430_, which binds to the Methyl-co-enzyme M reductase and catalyzes Methyl-S-co-enzyme M to methane in almost all methanogenic pathways [105]. The assistance of biochar was proven to decrease the dosage of trace elements. Cai et al. [106] found that the addition of biochar resulted in a 50% decrease in trace element demand when the OLR was 5 g TS/L day, with the carbon-carbon conversion rate slightly increasing from 26.2% to 29.5%.

In the mesophilic CSTR anaerobic digestion of wheat stillage, Schmidt et al. [110] found that a deficiency in Fe resulted in the accumulation of propionate, and the depletion of Ni could cause an increase in both acetate and propionate. When Osuna et al. [111] increased the OLR from 5 g COD/L day to 10 g COD/L day in UASB reactors with trace metal supplied (including Fe, Co, Ni, Mn, Cu, Zn, Mo, and Se) or not supplied, there were no discrepancies in acetate or butyrate degradation between the two reactors, but a significant difference was observed in propionate degradation. Propionate concentration in a trace metal-supplied trail showed fluctuations but regained a low level after 40 days of operation, which was contrary to continuous propionate accumulation in a trace metal-deprived reactor [111]. Daniel et al. [112] reported that the addition of either Fe, Ni, or Co or all three could increase the propionate utilization rates in mesophilic and thermophilic digesters by as much as 50%, with a more significant effect in the thermophilic systems. Ezebuiro et al. [113] confirmed that Co played an important role in propionate degradation. Supplementation using Co made the MMC pathway more energy-saving for micro-organisms and was thermodynamically more beneficial when compared to no Co supplementation. Propionate accumulation occurred easily in the anaerobic digestion of food waste due to a deficiency in Fi, Co, Ni, and Mo. In the long-term anaerobic digestion of food waste, propionate inhibition was identified as the main reason for failure in the anaerobic digestion of food waste, but this could be eliminated by Fe, Co, Ni, and Mo supplementation [114]. The simultaneous addition of the cheating agent [S,S]-EDDS could improve the soluble fractions of the metals by 1.2–7 times when compared to non-[S,S]-EDDS addition trails, with higher bioavailability and easier uptake by micro-organisms [105,114]. By using high-throughput sequencing analysis, Zhang et al. [115] revealed that in the long-term continuous anaerobic digestion of food waste, the absence of Fe, Co, Ni, and Mo resulted in a change in methanogenic community. In the seed sludge, *Methanosarcina*, which are reported to use acetate, hydrogen, and methyl compounds, was the dominant methanogen (73.5%). After a period of charging and discharging, the proportion of *Methanosarcina* gradually declined, accompanied by increasing *Methanosaeta*, which is a unique acetotrophic methanogen that replaced the dominant position of *Methanosarcina*. As a result, the hydrogen could not be consumed in a timely manner, and this caused propionate accumulation. However, when Fe, Co, Ni, and Mo was added, this effectively prevented the shift in the methanogenic community and maintained the dominant position of *Methanosarcina*. As a result, the carbon conversion rate of food waste using anaerobic digestion increased from 3.5% to 3.9% (at OLR of 3.0 g VS/L day).

Mo, W, and Se are also important in propionate degradation. Formate dehydrogenase (FDH), which is an essential enzyme in propionate-oxidizing bacteria for producing formate, contains Fe, Se, Mo, and W. Worm et al. [116] found that the long-term absence of Mo, W, and Se in the feed of a UASB reactor caused a decrease in specific methane activity with propionate as the substrate, which was due to the decrease in activity of *Syntrophobacter* sp. and the accumulation of a competitor. Banks et al. [117] reported that the absence of Se and Co resulted in the inhibition of propionate-oxidizing bacteria and formate-reducing hydrogenotrophic methanogens, which further caused a loss of syntrophic interspecies electron transfer and non-reversible propionate accumulation. Conversely, supplementation with Se and Co could prevent accumulation. Jiang et al. [118] comprehensively investigated the effect of the trace elements Se, Mo, Co, and Ni on the anaerobic digestion of food waste using a fractional factorial experimental design. They proved that Se played a key role in improving the degradation rates of acetic and propionic acid; Mo and Co had a modest effect on promoting propionate degradation; Ni showed a slight inhibitory effect on all VFAs conversion. Significant synergistic interactions were observed between the VFAs degradation rate (including propionate) and supplementary of Se or Mo by Ezebuiro et al. [113].

Regarding trace element interactions, Ezebuiro et al. [113] reported significant synergistic interactions between Ni and Co and Ni and Se, as well as antagonistic interactions between Co and Mo on VFAs degradation. The antagonistic effect of a combination of W/Mo on propionate degradation was found by Jiang et al. [118]. The combination of trace metals with other stimulation factors may have a 1 + 1 > 2 performance. Capson-Tojo et al. [119] were the first to report that simultaneously providing trace elements and granular carbon further enhanced the propionate degradation rate (0.37 g propionate/L day) compared to providing only trace elements (0.28 g propionate/L day) or granular carbon (0.24 g propionate/L day). Additionally, the propionate degradation duration time was shortened. The positive synergistic effect could be attributed to a decrease in thermodynamic energy when introducing granular carbon and the enhanced growth of propionate-degrading micro-organisms when introducing trace elements.

In general, supplementation using trace metals is essential for anaerobic microbes’ synthesis and activities. A suitable metal type and concentration could stimulate the microbes’ performance. However, considering the possible ecotoxicological risks of trace metals, much future research effort is still needed for reducing the trace metals supply dosage, such as using a heating agent, recycling the dosed trace metals, or replacing the trace metals via co-digestion with trace metal abundant waste (sewage sludge and animal waste). Since reactor configuration could influence the degradation of propionate, it may also affect the concentration of the trace metals required, yet information about this is still lacking. Moreover, the catalysis potential of trace elements in solid residues in subsequent pyrolysis treatments and immobilization performance still need to be explored.

### 4.4. Addition of Sulfate

In anaerobic digestion systems, both mutualistic and competitive interactions between sulfate-reducing bacteria (SRB) and methanogenic-producing bacteria (MPB) exist [120]. SRB, such as *Desulfobulbus propionicus*, *Desulfosporosinus*, *Desulfovibrio*-related SRBs, compete with BMP for substrates, such as hydrogen, formate, acetate, propionate, butyrate, primarily propionate, and hydrogen [121]. Due to the faster growth rate and higher affinity with propionate, the degradation of propionate combined with sulfate reduction via SRB is more favorable than the syntrophic oxidation of propionate [122]. The importance of SRB in the conversion of propionate and hydrogen has been reported by many researchers and was reviewed by Chen et al. [88].

The outcome of the competition between SRB and MPB was reported to be particularly determined by three factors: temperature, OLR, and SO_4_^2−^/COD. Generally, SRB could outperform MPB under mesophilic conditions, while MPB was more competitive in thermophilic systems [123]. SRB seem superior at low substrate levels (such as < 0.5 g COD/L) [124]. SO_4_^2−^/COD is an important factor that influences electron flow and strongly determines the winner in a competition. As the ratio increases, SRB will predominate, and MPB predominate as the ratio decreases. Choi et al. [125] reported that SRB and MPB were very competitive at an SO_4_^2−^/COD ratio of 0.37–0.59, above which range SRB outperformed MPB and below which MPB predominated. Many researchers had reported system deterioration when the SO_4_^2−^/COD ratio was higher than 0.1 [126,127,128]. In a lab-scale UASB reactor, Jiménez et al. [128] found that when the ratio of SO_4_^2−^/COD was 0.05, only 4.5 ± 0.3% COD removal was accomplished by SRB with no interference with methane production. Similarly, Erdirencelebi et al. [126] reported a value of 3% at an SO_4_^2−^/COD ratio of 0.05. Therefore, a low level of sulfate in an anaerobic digester is allowed without decreasing the activities of propionate-oxidizing bacteria and methane generation. When the SO_4_^2−^/COD ratio is below 1.88, the electron donors are not sufficient for SRBS, and, as a result, they are at a competitive disadvantage to MPBS [129].

Sulfate addition as a positive method to improve propionate degradation and methane production was reported by Li et al. [18]. In an anaerobic membrane reactor (AnMBR), when the OLR was increased gradually from the initial 3.9 kg COD/m^3^d to 14.6 kg COD/m^3^d, an inhibitory effect and a tendency toward deterioration was observed, with severe VFAs accumulation (2134 g COD/L) dominated by propionate (2070 g COD/L). Stop feeding and alkali supplements failed to recover the system, while the addition of Na_2_SO_4_ at 200 mg-S/L accelerated the conversion of propionate, and the subsequent decrease in H_2_ partial pressure further favored the degradation of propionate. Finally, the anaerobic system recovered after about 50 days, and stable operation at a higher OLR of 15.2 kg COD/m^3^d was achieved. The degradation of propionate before and after sulfate addition can be seen in Figure 3. In another piece of research [122], thermodynamic calculations were performed, and the results indicated that in a sulfate supplementation reactor, the H_2_ oxidation by carbonate (Environmental ΔG ≈ −95–100 kJ/mol) was more thermodynamically feasible than oxidation via sulfate (Environmental ΔG ≈ −20 kJ/mol), while propionate oxidation coupled with sulfate reduction (Environmental ΔG ≈ −190 kJ/mol) was more advantageous than the acetogenesis of propionate (Environmental ΔG ≈ −5–5 kJ/mol), which supported the idea that propionate in a high OLR (low HRT) system could be more effectively degraded by SRB by introducing sulfate, and supplementing sulfate in a high OLR system was a feasible alternative for enhancing propionate degradation and maintaining reactor performance. When considering the superior competitiveness of MBP in thermophilic digestion yet inferior competitiveness in mesophilic digestion when compared with SRB, the effectiveness of supplying sulfate for enhancing the degradation of propionate in mesophilic anaerobic digestion remains unproven. The long-term influence of introducing SRB and the possible evolution in the microbial community in a mesophilic or thermophilic digester is unknown. Furthermore, the feasibility of supplying sulfate-rich waste via co-digestion instead of pure sulfate needs more evidence.

### 4.5. Addition of Nitrogen-Containing Compound

Based on the principle that denitrification is an electron-accepting process, the coupling of denitrification and hydrogen consumption in an anaerobic digester was proposed by some researchers. Li et al. [130] investigated the effect of nitrate addition on propionate degradation in semi-continuous reactors. The results indicated that 130 mg/L of nitrate in a 1000 mg propionate/L reactor achieved higher propionate removal efficiency (74.7%) when compared with no nitrate dosing (68.5%). When increasing the nitrate dosing to 260 mg/L, the propionate removal efficiency rose further to 90.8%, suggesting that supplementation using nitrate could accelerate the degradation of propionate. However, the consumption of propionate by denitrification also resulted in a decrease in biogas yield, which was due to the denitrifying bacteria not only competing with methanogens for hydrogen but also they had a higher affinity with acetate, resulting in the removal of the partial methanogenic substrate. Thus, nitrate addition for enhancing propionate might be used transiently in anaerobic digesters for some specific purposes, such as the recovery of a deteriorated digester with high propionate accumulation or increasing the granular size of sludge. The possibility of adding an opportune nitrate to a digester to adjust hydrogen partial pressure with no or little effect on methane production still needs further evidence, and the effectiveness of co-digestion with nitrate-rich waste instead of pure nitrate reagent would add some value to this subject.

Another study by Li et al. [131] proposed the addition of azo dye into a wastewater anaerobic digester to accelerate VFAs decomposition and azo decolorization synchronously. When feeding the digester with a high propionate dose (1800 mg COD/L), the addition of azo at 35 mg/L achieved almost 80% propionate degradation, with a colorization rate of nearly 75%. However, higher azo dosing, such as 70 mg/L or 120 mg/L, harmed the anaerobic system, as reflected by the lower degradation rate of propionate and colorization rate of azo. When comparing the acetification process of propionate with or without azo addition, the researchers found that the moderate addition of azo (35 mg/L) accelerated the acetification process of propionate, with the highest conversion rate of propionate to acetate. Simultaneously, 0.002 mol/L hydrogen was observed in a sole propionate feeding system, while no hydrogen could be detected in a propionate + azo feeding system. This indicated that after the addition of azo dye, the hydrogen was utilized as an electron donor for the reduction of azo bonds and decolorization. The consumption of hydrogen in the system makes it more thermodynamically beneficial for propionate decomposition. The biological analysis also showed a higher abundance of propionate-utilizing acetogenic bacteria with azo dye addition to the digester when compared to the control. But the effect of azo dye addition on methane production was not reported. Although supplementation using sulfate or nitrate could effectively accelerate the conversion of propionate, according to electronic conservation, they sacrifice the total methane yield to some extent, which still needs consideration and further investigation.

### 4.6. Addition of Conductive Material

The long-standing mechanism described for interspecies electron transfer among syntrophic consortia is hydrogen and formate transfer. Hydrogen transfer is preferable when the interbacterial spatial distances are short, while the formate route is dominant when the distances are long. About 10 years ago, however, researchers found that direct interspecies electron transfer (DIET) played an important role in electron exchange, and several lines of evidence were presented [49,132]. Electrical connections and electrons shuttled among the syntrophic consortia were directly accomplished by electrically conductive pili and the associated c-type cytochrome [133,134]. Thus, various kinds of conductive materials, like activated carbon [135], graphene [136], magnetite [137,138], and biochar [139,140], were provided in methanogenic environments in order to stimulate DIET, as well as the organic conversion rate. The main mechanism is that the presence of conductive engineering material can substitute conductive pili and c-type cytochrome with inherent conductivity.

In 2011, Masahiko et al. [141] first reported the electrical conductivity of methanogenic aggregates derived from a UASB reactor treating brewery wastewater. Microbial community analysis via 16S rRNA gene sequencing suggested that the DIET between *Geobacter species* ad *Methanosaeta concilii* may have contributed to the enhancement of the degradation of ethanol. Since then, intensive works have been conducted to demonstrate the existence of DIET in ethanol anaerobic biodegradation via co-culture experiments using *Methanosarcina barkeri* with *Geobacter metallireducens* [142], *Geobacter metallireducens* with *Methanosaeta harundinacea* [143], *Geobacter metallireducens* with *Geobacter sulfurreducens* [142], or mixed microbial communities [138]. Although there are some studies that have used a complex substrate like sewage sludge, municipal solid waste leachate, food waste, or dog food, few studies have been reported, and little evidence on propionate metabolism via DIET has been provided (Table 2). Theoretically, the degradation of propionate through DIET is expected to follow this equation: CH_3_CH_3_COO^+^ +3 H_2_O **→** CH_3_COO^+^ +HCO_3_^−^ + 7H^+^ + 6e^−^ (ΔG0 = −189.7 kJ/mol, at 25 °C). When compared with syntrophic degradation coupled with H_2_ consumption (equation (i + M), Table 1), the DIET pathway has a lower Gibbs free energy, indicating that it is more thermodynamically favorable [132].

Yamada et al. [154] observed that it took more than 150 days to completely degrade 25 mM propionate in a thermophilic reactor, while supplementation using magnetite or ferrihydrite could shorten the degradation time to less than 50 days. The promotion mechanism was speculated to occur due to the enhancement of DIET, but the micro-organisms involved during propionate degradation were not identified. In a semi-continuous experiment using propionate as the substrate and supplying magnetite (20 mM), Yang et al. [64] obtained a culture greatly enriched with *Thauera*, which was reported to be probably capable of DIET, and *Methanobacterium* that could utilize H_2_ to produce methane, indicating the simultaneous occurrence of propionate degradation in the DIET and H_2_ transfer paths. Zhao et al. [139] used biochar as an electron conduit to enhance DIET for the syntrophic metabolism of propionate in an upflow anaerobic sludge blanket reactor. The results showed that the addition of biochar enhanced the propionate removal efficiency from 87.8% to 98%, with the *Geobacter* and *Methanosaeta species* greatly enriched, suggesting the possibility of propionate metabolism via DIET. Another study by Zhao et al. [149] revealed that without conductive material supplementation, a period of ethanol addition in a UASB reactor using propionate as the substrate could significantly enrich *Gepbacter*, *Methanosaeta,* and *Methanosarcina* species, which were reported as syntrophic partners and possibly capable of DIET, while in traditional propionate domestication enriched culture, H_2_-utilizing methanogens (such as *Methanolinea species*) were enriched and aceticlastic methanogens, such as the *Methanosaeta* and *Methanosarcina* species, were 18.74% lower than in a ethanol-propionate-stimulated reactor. The ethanol-propionate-stimulated enrichment could resist higher OLRs and hydrogen partial pressure stresses, which might be due to the establishment of DIET and enhanced syntrophic metabolism. These results indicated that the syntrophic capabilities of DIET involved in ethanol metabolism might also be capable of propionate conversion via DIET. With supplementation using Fe oxide-loaded carbon cloth, Xu et al. [147] revealed that the propionate degradation and cumulative CH_4_ production increased by 19.67% and 15.4% compared with the control.

More direct evidence was provided by proteomic analysis [148]. The results suggested that the addition of magnetite directly induced changes in protein expression levels in propionate conversion, which was attributed to the changed microbial species that had a different, specific metabolic pathway and a different number of proteins. As a result, 11 enzymes, including methylmalonyl-CoA, succinyl-CoA, and acetyl-CoA, which might originate from some known propionate-oxidizing bacteria, such as *Pelotomaculum*, *Syntrophobacter*, were upregulated. The cytochrome c oxidase-related protein that originated from *Thauera* was upregulated, which might be associated with DIET, but this still needs further evidence. In contrast to other research, the upregulated proteins originating from *Geobacteraceae* were not found in their study, which might represent indirect evidence of *Geobacteraceae* not participating in propionate metabolism or the DIET process.

A recent study by Walker et al. [155] first reported that *S. aciditrophicus*, which is outside of the genus *Geobater*, could grow via DIET. They speculated that DIET was a likely option for microbes when re-examining HIT. Thus, providing conditions that favor propionate degradation through DIET would need further investigation.

Although the co-culture of *Methanosarcina barkeri* with *Geobacter metallireducens* was previously reported to metabolize ethanol through the DIET route [142], Wang et al. [156] found they could not metabolize propionate as a sole electron donor. Likewise, the co-culture of *G. metallireducens* and *Methanosaeta harundinacea*, which were proven to be capable of metabolizing ethanol through the DIET route [143], also could not metabolize propionate when propionate was used as the sole electron donor [156]. The co-culture results by Wang et al. [156] seem somewhat contradictory to those of Zhao et al. [149], who thought that ethanol addition stimulated DIET during propionate metabolism in mixed microbial communities. Therefore, more efforts to explore co-culture experiments to define micro-organisms using propionate as the sole electron donor, as well as establishing an electrical path like microbial fuel cells (MFCs), and the isolation and identification of micro-organisms capable of metabolizing propionate via DIET from mixed syntrophic consortia, are required for further validation of propionate metabolism through DIET.

### 4.7. Nano-Sized Additives

In general, trace metal supplementations in AD for enhancing propionate degradation are in the form of a dissolved state. Recently, more researchers have started to pay attention to adding nano-sized trace metals for enhancing VFAs conversion and methane generation since nano-sized particles have their unique small-object effects [157,158,159]. Tian et al. [160] found that although 400 mg/L of nano-sized MnO_2_ could vastly inhibit the conversion of propionate, a proper concentration (50 mg/L) could stimulate the methanogenesis of propionate due to it triggering the stress response of anaerobic digester sludge and causing more enzymes to be secreted and participate in the digestion reactions. When using propionate as the sole substrate, Jing et al. [148] revealed that supplementation using 10 mg/L nano-magnetite could stimulate methane production by 44% in batch experiments. In the experiment of using acid-resistant inoculation sludge as the fermentation substrate to explore the influence of nano-magnetite on the anaerobic digestion system, the propionic acid degradation rate of the nano-magnetite group was 10.96–74.62% higher than that of the control group without nano-magnetite [161]. In waste cooking oil and aerobic sludge fermentation systems, when the added concentration of nano-Fe_3_O_4_ was 5 g/L, the microbial community evolved in a direction conducive to the production of propionic acid [162]. Under this condition, the concentration of propionic acid reached 4990.92 ± 124.76 mg COD/L. Nano-magnetite-enriched culture was proven to have higher H_2_-utilizing activity and a higher abundance of *Thauera*, which was reported to be linked with direct interspecies electron transfer, indicating that both modes of propionate (HIT and DIET) were stimulated with nano-magnetite addition [148]. Due to its reductive characteristics and playing the role of a trace metal, nano-sized zero-valent iron (NZVI) could ensure a low OPR and prevent propionate generation [62].

### 4.8. Degasification of Dissolved Hydrogen

Degasification via membranes in anaerobic digestion was commonly used for to recover dissolved CH_4_, adsorb CO_2_, upgrade the biogas [163,164,165], or separate hydrogen from hydrogen in a production digester. Few studies could be found that used a membrane to directly separate the dissolved hydrogen from the digester. Satoh et al. [65] proposed using a hollow fiber degassing membrane (DM) to remove the dissolved hydrogen gas concentration so as to enhance propionate and acetate degradation and methane production. The test in two bench-scale reactors showed that the dissolved hydrogen in a liquid phase greatly decreased after applying the DM. At a high OLR of 122.9 g COD/L day, the propionate concentration in the control reactor increased to 500 mg/L and showed an accumulation trend, while the DM reactor could still maintain at low level, lower than 100 mg/L. This was due to the favorable thermodynamics of propionate degradation after hydrogen removal in the DM reactor. The methane production in the DM reactor was 20% higher than in the control reactor, which indicated hydrogen loss by degasification did not reduce methanogenesis.

## 5. Conclusions and Perspectives

Propionate inhibition exists ubiquitously in anaerobic digestion processes, and its degradation is a critical step to substrate conversion. Propionate oxidation has the highest Gibbs free energy compared to other VFAs, and it requires syntrophic co-operation between POBs and methanogenic archaea. However, understanding the propionate limits for various digester configurations, and the impacts of temperature, the ORP, and adapted microbial communities needs to be probed further. A clear elucidation of the pathways of propionate generation and biodegradation is required to provide guidance when adjusting and optimizing the parameters. When compared to mesophilic anaerobic digestion, thermophilic anaerobic digestion suffers from worse operation stability and is more susceptible to propionate accumulation and inhibition; thus, the application of strategies in thermophilic digesters is more challenging.

Bioaugmentation has been adopted successfully in batch experiments or in continuous experiments for rescuing propionate inhibition reactors, but it is still challenging to apply it in continuous reactors because the functional micro-organisms are easily washed out and may need long-term supplementation. How to make the micro-organisms really participate in the establishment of anaerobic flora or immobilize them in reactors requires more research. Previous investigations demonstrated the efficacy of the fed-batch biochar-amended AD system, while further investigation efforts should be paid to its application in a continuous/semi-continuous anaerobic process and its recycling and reuse. In addition, prudent procedures should be developed to avoid the adverse effects exhibited by excessive biochar dosage, and desirable biochar synthetic conditions should be specified to make the biochar-amended AD process more productive and cost-effective.

There have been various strategies proposed to enhance the degradation of propionate; however, the mechanisms of some of the regulation methods, such as the addition of conductive materials, are not completely understood and need further in-depth studies. It is worthwhile to point out that the numbers of full-scale demonstration trials are very small and requires more attempts. Full-scale applications present more complex reaction procedures. The properties of the additives in anaerobic digestion reactors exhibit a good correlation with the biodegradation enhancement of propionate, but the mechanisms of functional microbial metabolic routes in conductive material-amended digesters are not yet understood, and more efforts should be paid towards this. Besides, techno-economic analyses and comparisons between the different strategies in full-scale demonstrations, together with mass and energy balance evaluations, are essential. The selection and application of methods for enhancing propionate biodegradation, and application-oriented material production and addition, and the associated energy consumption should be justified from the perspective of the life cycle.

## Figures and Tables

**Figure 1 molecules-28-03883-f001:**
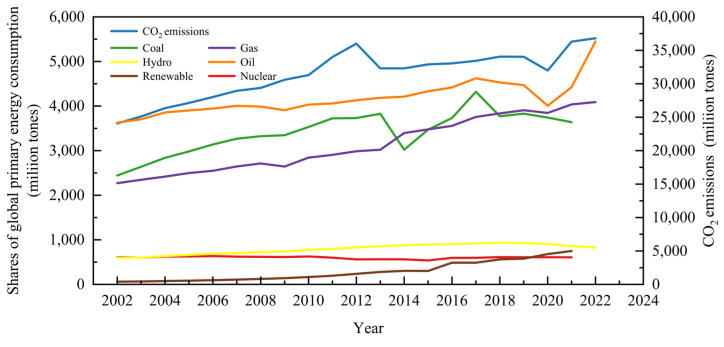
Global primary energy consumption by fuel sources and CO_2_ emissions.

**Figure 2 molecules-28-03883-f002:**
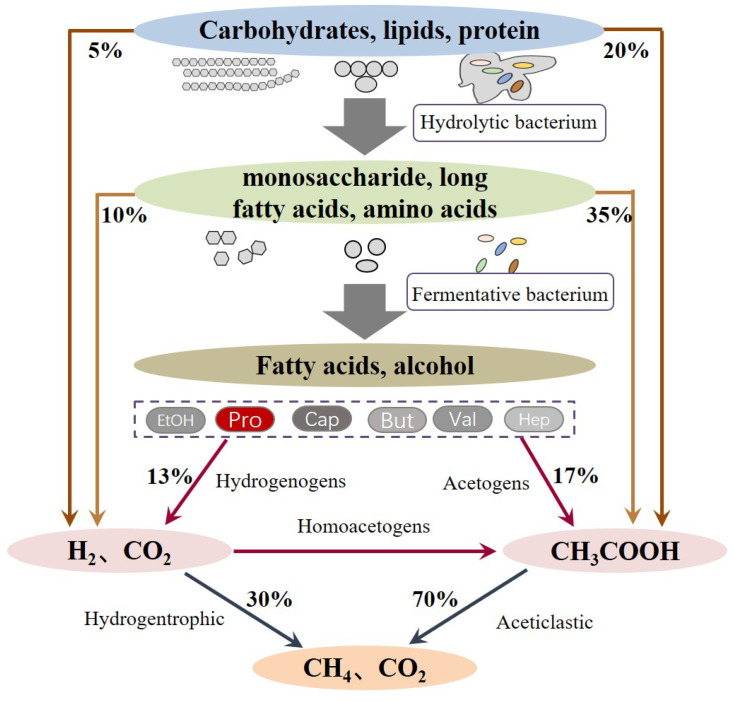
Stages and main pathways in anaerobic digestion.

**Figure 3 molecules-28-03883-f003:**
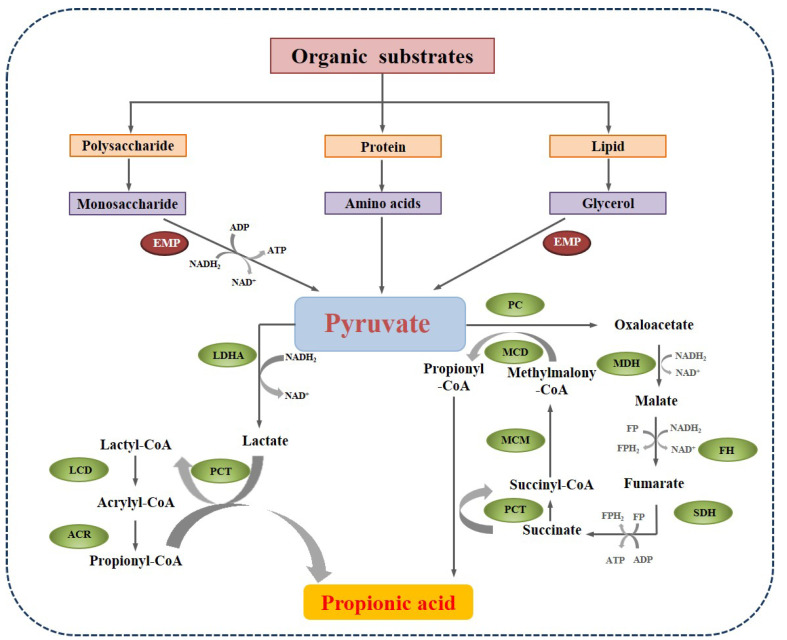
The main pathways of propionate formation in anaerobic digestion (EMP: Embden-Meyerhof-Parnas pathway; LDHA: lactate dehydrogenase; PCT: propionyl-CoA transferase; LCD: lactyl-CoA dehydratase; ACR: acrylyl-CoA reductase; MDH: malate degydrogenase; FH: fumarate hydratase; SDH: succinate dehydrogenase; MCM: methylmalonyl-CoA reductase).

**Figure 4 molecules-28-03883-f004:**
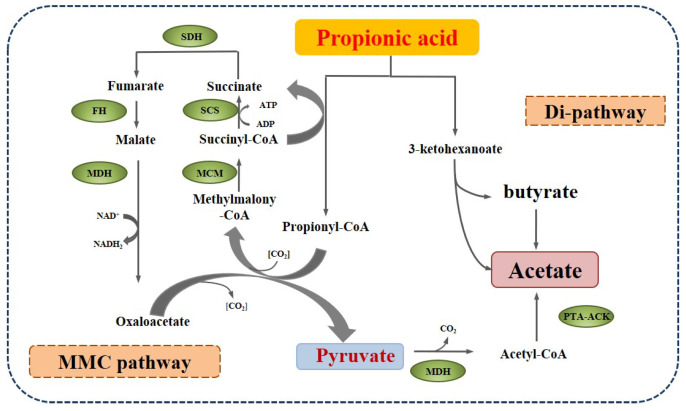
The main degradation pathways of propionate in anaerobic digestion (PTA-ACK: phosphotransacetylase-acetate kinase).

**Figure 5 molecules-28-03883-f005:**
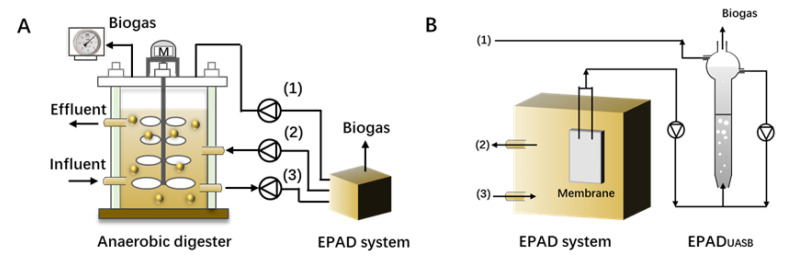
(**A**) EPAD system connected to a full-scale anaerobic digester; (**B**) detailed flow sheet of EPAD system (adapted with permission from Ref. [103], (1)-biogas production; (2)-effluent of EPAD system; (3)-influent of EPAD system).

**Table 1 molecules-28-03883-t001:** Degradation reaction of fatty acids and the standard Gibbs free energy.

Reaction	ΔG (kJ/mol, 25 °C)
(i) CH_3_COO^−^ + H_2_O → HCO_3_^−^ + CH_4_	−31.0
(ii) CH_3_CH_2_COO^−^ + 3 H_2_O → CH_3_COO^−^ + HCO_3_^−^ + H^+^ + 3 H_2_	+76.1
(iii) CH_3_CH_2_CH_2_COO^−^ + 2H_2_O → 2CH_3_COO^−^ + H^+^ + 2H_2_	+48.4
(iv) CH_3_CH_2_CH_2_CH_2_COO^−^ + 2H_2_O → CH_3_COO^−^ + CH_3_CH_2_COO^−^ + H^+^ + 2H_2_	+25.1
(M) 4 H_2_ + HCO_3_^−^ + H^+^ → CH_4_ + 3 H_2_O	−135.6
(ii + M) 4 CH_3_CH_2_COO^−^ + 3H_2_O → 4 CH_3_COO^−^ + HCO_3_^−^ + H+ + 3 CH_4_	−102.4

**Table 2 molecules-28-03883-t002:** Summary of reported methanogenic communities using propionate as an electron donor.

Possible Electrophilic Micro-Organism or Electron Acceptor	Possible Electron Donator	Conduit	Employed Concentration	References
*Methanothrix*	*Syntrophobacteraceae Thiobacillaceae*	Pyrite	5–40 g/L	[144]
*Syntrophobacter fumaroxidans*	*Geobacter sulfurreducens*	electric wire	/	[145]
*Geobacter*, *Syntrophobacter*, *Smithella*, and *Methanosaeta*	*Geobacter*	coupled effects of ethanol and Fe_3_O_4_	500 mg COD/L ethanol + 10 g/L Fe_3_O_4_	[146]
*Methanothrix*	*Levilinea*	Fe_2_O_3_-loaded carbon cloth	/	[147]
*Methanospirillum*, *Methanosphaerula*	*Thauera* sp.	Magnetite	10–1000 mg/L	[148]
*Methanosaeta* sp.	*Geobacter* sp.	biochar	5 g/L	[149]
*Methanosaeta* sp., *Methanosarcina* sp.	*Geobacter* sp.	Electronically conductive pili		[149]
*CO_2_-reducing methanogens*	*Propionate-oxidazing acetogens*	Magnetite		[150]
*Methanobacterium*	*Thauera* sp.	Magnetite	20 mM	[64]
*Methanosaeta* and *Methanosarcina* sp.	*Geobacter*	Graphite felt		[151]
*Methanobacterium* sp.	*Anaerolineae* and *Clostridia*	Granular activated carbon	0∼5.0 g	[152]
*Methanosaeta* sp.	*Propionate* and *Butyrate*	carbon fibers		[153]

## Data Availability

No new data were created or analyzed in this study. Data sharing is not applicable to this article.

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
