# Peer review of "Emerging Strategies for Enhancing Propionate Conversion in Anaerobic Digestion: A Review"

_molecules, 2023, doi:10.3390/molecules28093883_

Round 1

Reviewer 1 Report

Dear authors, after carefully reading the article, I have suggestions for its significant improvement.

1. The article would have been significantly improved if the authors had given the dynamics of the construction of anaerobic digestion plants over the past 10-20 years.

2. The quality of Fig.1  needs to be improved.

3. The review does not touch at all on the kinetic of anaerobic digestion reactions.

4. There is no solution to the problem of solid residue and its quality (ash content, fixed carbon content, etc.).

5. The authors did not indicate how the yield, composition and heat of combustion of gas change depending on temperature, pH, oxidation-reduction potential, acetate concentration and H2 pressure, propionate concentration and other factors. Section 3 is very poor.

6. It is necessary to add various technological schemes of anaerobic digestion actually introduced into production.

In general, the article needs significant improvement.

Author Response

Overall comments: 

Dear authors, after carefully reading the article, I have suggestions for its significant improvement.

General comments:

Point 1: The article would have been significantly improved if the authors had given the dynamics of the construction of anaerobic digestion plants over the past 10-20 years.

Response 1: Thanks for reviewer’s valuable suggestion. In the introduction, the constructions of anaerobic digestors and the development of anaerobic digestion plants over the past 10-20 years was summarized.

  • On Page 2 of the revised manuscript:

In recent years, more and more countries and regions have begun to widely use this technology for the production of bioenergy. For instance, Europe counted over 18943 commercial combined heat and power (CHP) plants in 2019. They totally pro-duced 167 TWh biogas for CHP utilization and 26 TWh purified methane for injection into the municipal grid. Germany has more than 8000 AD plants, which generates ap-proximately 4.0*1010 kWh/year [8]; Australia intends to achieve 5.6*1010 kWh/year bioenergy generation through AD plants in 2050 [9]. In USA, the government provides AD plants with a 1.1ï¿ /kWh tax credit for the first ten years for supporting the develop-ment of AD technology [10]. Anaerobic digestion treatment of organic waste is prevail-ing in recent years. Cambi® thermal hydrolysis + anaerobic digestion developed fast and has quickly taken over the market in the past 10 years. The installed AD plants in-clude Blue Plains wastewater treatment plants in USA, Chertsey, Riverside, Crawley, Beckton, Crossness, Long Reach and Oxford wastewater treatment plants in UK, Xiaohongmen, Gaobeidian, gao’antun in China and so on. In addition, co-digestion of multiple substrates which can compensate for disadvantages in single digestion was adopted in many cases [122]. Although AD plants installed in developing countries (Chi-na, India, Brazil) are mainly for treatment of municipal sewage sludge, it is expected to growth rapidly in the next few decades [11].

Point 2: The quality of Fig.1 needs to be improved

Response 2: Thanks to the reviewer's comment. The quality of Fig.1 and other figures was improved in the revised manuscript.

  • On Page 2 of the revised manuscript:

Figure 1. Global primary energy consumption by fuel sources and CO2 emissions

  • On Page 3 of the revised manuscript:

Figure 2. Stages and main pathways in anaerobic digestion

  • On Page 4 of the revised manuscript:

Figure 3. The main pathways of propionate formation in anaerobic digestion (EMP: Embden-Meyerhof-Parnas pathway; LDHA: Lactate dehydrogenase; PCT:Propionyl-CoA transferase; LCD: Lactyl-CoA dehydratase; ACR: acrylyl-CoA reductase; MDH:malate degydrogenase; FH:fumarate hydratase; SDH:succinate dehydrogenase; MCM: methylmalonyl-CoA reductase)

  • On Page 5 of the revised manuscript:

Figure 4. The main degradation pathways of propionate in anaerobic digestion (PTA-ACK: phosphotransacetylase-acetate kinase)

Point 3: The review does not touch at all on the kinetic of anaerobic digestion reactions.

Response 3: Thank you for your valuable advice. Contents about kinetic of anaerobic digestion reactions were added in the revised manuscript, including the modifications of conventional kinetics and the developed machine learning coupled kinetics.

  • On Pages 7-8 of the revised manuscript:

An appropriate kinetic model can analyze the changes in the process of anaerobic digestion reaction, which is conducive to the accurate grasp of the reaction results, im-proving the stability of the system, and providing guidance for the operation and con-trol of anaerobic digestion. At present, dynamic models have been studied and applied as follows: first order kinetic model [69], modified Gompertz model [70], the anaerobic di-gestion model No.1 (ADM1) [71], two-phase exponential model [72], and multi-stage [72]. As a conventional model, the first order dynamic model is used to depict the biogas yield process of different biomass wastes [73, 74]. The formation of biogas is related with the activity of methanogens during anaerobic digestion, thus the modified Gompertz model which is derived from the growth model of mixed populations was considered as a good non-linear regression model [75]. The fitting error of the theoretical and practical results of the first order model is larger than that of the theoretical and experimental results of the modified Gompertz model [76]. The Modified Gompertz model can be used to obtain the retention time of biogas generation and the maximum methane yield. ADM1, another commonly used kinetic models, makes the fitting precision between the experimental results and the simulated values smaller, which gives the simulated concentrations of soluble organic components [77].

Existing studies have found that the superposition model for co-digestion of different biomass can more accurately predict the potential of methane production. Wang et al [80] The superposition model was used to obtain a good fit for the co-digestion of pig manure and kitchen waste (R2=0.99). Adarme et al (2023) [81] proposed the two-phase exponential model, which is also used to describe methane production. In recent years, machine learning model was introduced to accurately predict the kinetic parameters in anaerobic digestion model. In a previous study, Ge et al (2023) [78] found that after model optimization, the average R2 for predicting 7 kinetic parameters, including disin-tegration constant (Kdis), hydrolysis constant of carbohydrates (Khyd-CH), half-saturation constants of acetate (Ks_ac), half-saturation constants of monosaccharide (Ks_su) etc., reached 0.92, and the root mean square error reached 0.167.

  • On Page 5, Line 154-155 of the revised manuscript:

The Gibbs free energy of propionate oxidation is +76.1 kJ/mol, which is thermodynami-cally unfavorable [45].

Point 4: There is no solution to the problem of solid residue and its quality (ash content, fixed carbon content, etc.).

Response 4: Thanks to the reviewer's remind. According to reviewer’s suggestion, some discuss and expectation about the solid residue treatment was supplemented in the revised manuscript.

  • On Page 2, Lines 57-61 of the revised manuscript:

The waste residue generated by its fermentation contains a large amount of trace metal elements and nutrients that can stimulate the growth and development of plants, so it can also be used as organic fertilizers; the derived biochar can be recycled to regulate the stability of the anaerobic digestive system [14].

  • On Page 8, Lines 342-345 of the revised manuscript:

In addition, supplementation of NaHCO3 or NaOH at the front end may lead to a high Na content solid residue and ash content, which is difficult to deal with.

  • On Page 12, Lines 496-498 of the revised manuscript:

Moreover, the catalysis potential of trace elements in the solid residue in subsequent pyrolysis treatment, and immobilizing performance need more explores.  

Point 5: The authors did not indicate how the yield, composition and heat of combustion of gas change depending on temperature, pH, oxidation-reduction potential, acetate concentration and H2 pressure, propionate concentration and other factors. Section 3 is very poor.

Response 5: Thanks very much for reviewer’s comment. According to the reviewer's suggestion, the Section 3 of the manuscript was revised.

  • On Page 6, Lines 203-204 of the revised manuscript

At the organic load rate (OLR) of 10.0 kg·COD/(m3·d), the methane production is 310 mL/g·COD.

  • On Page 6, Lines 208-212 of the revised manuscript

Zhao et al [55] investigated the effect of propionic acid on the activity of methanogene-sis. When the concentration of propionic acid is higher than 5000 mg/L, the AD system is completely inhibited. And compared with mesophilic digestor, the propionic acid degradation rate of the thermophilic digestor is relatively higher, with …

  • On Page 7, Lines 218-228 of the revised manuscript

Zhang et al. [58] used 2000 mg/L propionic acid as the only carbon source in an UASB re-actor. When the pH value was controlled at 6.8-7.5, the high propionate removal of the sludge blanket (81.5-90%) was observed, and the methane content and biogas produc-tivity were maintained at 55.2%-69.3% and 22.8-26.7 L/day; when pH was controlled at 6.0, 5.5 and 5.0, only 49.1%, 33.8% and 16.7% of propionate was degraded, the methane content decreased by 5.1% and 33.7% respectively, and the biogas productivity de-creased to 15.3 L/day. When the pH value is below 4.5, propionate degradation almost didn’t occur, and biogas production is almost stopped. The decline of the oxidation performance of propionate was related to the significant reduction of propionate oxi-dation bacteria and acetoclastic methanogens, which were more sensitive to low pH [58].

  • On Page 7, Lines 233-240 of the revised manuscript

The optimal ORP range for acid-producing fermentation is between -100 mV and -300 mV. The formation of propionate occurs under ORP conditions above -278 mV [60]. In an anaerobic reactor with iron added, the negative value of ORP is greater. If ORP is reduced to -300mV, it will inhibit the formation of propionate and increase the production of CH4. Therefore, ensuring the reduction atmosphere is critical to prevent the propionate fermentation route. Reducing additives, such as zero-valent iron, can be added to anaerobic digesters to prevent the formation of propionate and accelerate its degradation [61].

  • On Pages 7, Lines 246-252 of the revised manuscript

In Cazier's research, the influence of H2 and CO2 pressure separation on solid state AD was explored[64]. When the H2 partial pressure was increased from 0 to 600 mbar, the CH4 yield increased from 23±4 mL CH4/g TS to 28±6 mL CH4/g TS. When further in-crease the H2 partial pressure to 1555 mbars, the methane yield of CH4 decreased from 28±6 mL of CH4/g TS to 9±1 mL of CH4/g TS. Specially, when H2 is higher than 800-900 mbar, methane production decreased very fast.

  • On Pages 7, Lines 265-269 of the revised manuscript

Rafika et al [67] researched the performance of a semi-continuous anaerobic digestor under different OLRs. When OLR increased from 3.44 g VS/L d to 14.6 g VS/L·d, the methane productivity presented earlier increase and later decrease trend. Under the OLR of 4.25 gVS/L d, digestor showed a high energy potential of 530 L CH4/kg VS, while further increase of OLR lead to propionate accumulation.

  • On Pages 7-8, Lines 281-308 of the revised manuscript

An appropriate kinetic model can analyze the changes in the process of anaerobic digestion reaction, which is conducive to the accurate grasp of the reaction results, im-proving the stability of the system, and providing guidance for the operation and con-trol of anaerobic digestion. At present, dynamic models have been studied and applied as follows: first order kinetic model[69], modified Gompertz model[70], the anaerobic di-gestion model No.1 (ADM1)[71], two-phase exponential model[72], and multi-stage [72]. As a conventional model, the first order dynamic model is used to depict the biogas yield process of different biomass wastes[73, 74]. The formation of biogas is related with the activity of methanogens during anaerobic digestion, thus the modified Gompertz model which is derived from the growth model of mixed populations was considered as a good non-linear regression model[75]. The fitting error of the theoretical and practical results of the first order model is larger than that of the theoretical and experimental results of the modified Gompertz model [76].The Modified Gompertz model can be used to obtain the retention time of biogas generation and the maximum methane yield. ADM1, another commonly used kinetic models, makes the fitting precision between the experimental results and the simulated values smaller, which gives the simulated concentrations of soluble organic components[77].

Existing studies have found that the superposition model for co-digestion of dif-ferent biomass can more accurately predict the potential of methane production. Wang et al. [80] The superposition model was used to obtain a good fit for the co-digestion of pig manure and kitchen waste (R2=0.99). Adarme et al. [81] proposed the two-phase exponential model, which is also used to describe methane production. In recent years, machine learning model was introduced to accurately predict the kinetic parameters in anaerobic digestion model. In a previous study, Ge et al. [78] found that after model optimization, the average R2 for predicting 7 kinetic parameters, including disin-tegration constant (Kdis), hydrolysis constant of carbohydrates (Khyd-CH), half-saturation constants of acetate (Ks_ac), half-saturation constants of monosaccharide (Ks_su) etc., reached 0.92, and the root mean square error reached 0.167.

Point 6: It is necessary to add various technological schemes of anaerobic digestion actually introduced into production.

Response 6: Thanks for your valuable suggestion. Considering this manuscript was more concentrated on biodegradation of propionate and engineering strategies for propionate biodegradation, and detailed technological schemes of anaerobic digestion was reviewed by some other researchers, which may not the focus of this manuscript, a brief introduction of the commonly used anaerobic digestion schemes in recent years is added to the revised manuscript.

  • On Page 2, Lines 50-56 of the revised manuscript

…according to the form of fermentation tank, it can be divided into continuous stirred tank reactor (CSTR)[8], upflow anaerobic sludge bed(UASB)[9], internal circulation (IC)[10], expanded granular sludge bed (EGSB)[11], upflow solid reactor (USR)[12], etc.; according to the temperature of anaerobic fermentation, it can be divided into thermophilic, mesophilic and psychrophilic digestion. It is a relatively energy-saving and efficient treatment method. Carbon emissions from composting, landfill and waste incineration range from 61 to 1010 kg CO2-eq/t FW, which is much higher than AD. In addition,the biogas generated (50-75% CH4 and 25-45% CO2)[13] can be used for heat pro-duction, power generation or purification of natural gas; the waste residue generated by its fermentation contains a large amount of trace metal elements and nutrients that can stimulate the growth and development of plants, so it can also be used as organic fertilizers[14]; the derived biochar can be recycled to regulate the stability of the anaerobic digestive system, which were known as biochar-amended digestors [14].

Reviewer 2 Report

Comments: Abstract: Reframe the line 12-13 with deletion of etc. just mention VFA.

Line 17- delete etc and reframe.

Line 21-26: The statement is vague and english is unclear. Reframe and rewrite the portion.

Introduction needs more references! Please add relevant literature reviews.

Fig:1: Legend is unclear. Resolution should be increased. Choose a different display pattern.

Fig.3 and 4.Legend is unclear. Resolution should be increased.

Table 1: Scientific names should be in italics.

Instead of piling on the efficacies on AD, authors should discuss about the cumulative carbon conversion efficiency at the end of the process. the manuscript should focus on this aspect only and cite relevant literature.

Again, quality of english should be checked by a native speaker. Its not upto the required journal standard.

The quality of english should be checked by a native speaker. Its not upto the required journal standard.

Author Response

Overall comments

General Comments

Point 1: Abstract: Reframe the line 12-13 with deletion of etc. just mention VFA.

Response 1: Thanks for the suggestion. The sentence reviewer mentioned was modified in the revised manuscript.

  • On Page 1, Lines 12-16 of the revised manuscript

In the process of anaerobic digestion, pH, temperature, organic load, ammonia nitrogen, VFAs and other factors will affect the fermentation efficiency and stability of the system. The balance between the generation and consumption of volatile fatty acids (VFAs) as an important inter-mediate in the anaerobic digestion process has also become the key to restricting the stable operation of AD.

Point 2: Abstract: Line 17- delete etc and reframe.

Response 2: Thanks for review’s kind suggestion. The etc. on line 17 of the original manuscript was deleted and the sentence was reframed.

  • On Page 1, Lines 18-19 of the revised manuscript

To solve this problem, some strategies, including buffering addition, suspension of feeding, de-crease of organic loading rate and so on, has been proposed and developed.

Point 3: Line 21-26: Abstract: The statement is vague and english is unclear. Reframe and rewrite the portion.

Response 3: Thanks for your comments and suggestions. Abstract was reframed and rewrite, and the whole manuscript was checked and modified.

  • On Page 1 the revised manuscript

Anaerobic digestion (AD) is a triple-benefit biotechnology for organic waste treatment, renewable production and carbon emission reduction. In the process of anaerobic digestion, pH, temperature, organic load, ammonia nitrogen, VFAs and other factors will affect the fermentation efficiency and stability. The balance between the generation and consumption of volatile fatty acids (VFAs) in the anaerobic digestion process is the key to stable operation of AD. However, accumulation of VFAs frequently occurs, especially propionate, because its oxidation has the highest Gibbes free energy compared with other VFAs. To solve this problem, some strategies, including buffering addition, suspension of feeding, decrease of organic loading rate and so on, has been proposed. Emerging methods such as bioaugmentation, supplementary of trace elements, addition of electronic receptor, conductive materials and degasification of dissolved hydrogen, and so on, have been recently researched, presenting promising results. But the efficacy of these methods still requires further studies and tests in full-scale applications. The main objective of this paper is to provide a comprehensive review on the mechanism of propionate generation, metabolic pathways and the influencing factors during AD process, and the recent literatures on the experimental researches related to the efficacy of various strategies for enhancing propionate biodegradation. In addition, the issues that must be addressed in the future and focuses of future research were identified, and potential future development directions were predicted.

Point 4: Introduction needs more references! Please add relevant literature reviews.

Response 4: Thanks for your comment and suggestion. Some relevant literatures, including reviews, were added to the revised manuscript.

The additional literatures including:

  1. Pan, X., et al., Performance on a novel rotating bioreactor for dry anaerobic digestion: Efficiency and biological mechanism compared with wet fermentation. Energy, 2022. 254.
  2. Wang, Y., et al., Insight into the effects and mechanism of cellulose and hemicellulose on anaerobic digestion in a CSTR-AnMBR system during swine wastewater treatment. Sci Total Environ, 2023. 869: p. 161776.
  3. Lima, V.O., et al., Anaerobic digestion of vinasse and water treatment plant sludge increases methane production and stability of UASB reactors. J Environ Manage, 2023. 327: p. 116451.
  4. He, H., L. Liu, and H. Ma, The key regulative parameters in pilot-scale IC reactor for effective incineration landfill leachate treatment: Focus on the process performance and microbial community. Journal of Water Process Engineering, 2023. 51.
  5. Mortezaei, Y., T. Amani, and S. Elyasi, Corrigendum to ’High-rate anaerobic digestion of yogurt wastewater in a hybrid EGSB and fixed-bed reactor: Optimizing through response surface methodology’ [Process 113 (2018) 255–263]. Process Safety and Environmental Protection, 2023. 171.
  6. Yang, H., et al., Comparison of three biomass-retaining reactors of the ASBR, the UBF and the USR treating swine wastewater for biogas production. Renewable Energy, 2019. 138: p. 521-530.
  7. Veiga, S., et al., Biogas dry reforming over Ni-La-Ti catalysts for synthesis gas production: Effects of preparation method and biogas composition. Fuel, 2023. 346.
  8. Grandas Tavera, C., T. Raab, and L. Holguin Trujillo, Valorization of biogas digestate as organic fertilizer for closing the loop on the economic viability to develop biogas projects in Colombia. Cleaner and Circular Bioeconomy, 2023. 4.
  9. Perez-Esteban, N., et al., Potential of anaerobic co-fermentation in wastewater treatments plants: A review. Sci Total Environ, 2022. 813: p. 152498.
  10. Paranjpe, A., S. Saxena, and P. Jain, A review on pperformance improvement of anaerobic digestion using co-digestion of food waste and sewage sludge. J Environ Manage, 2023. 338: p. 117733.
  11. Ajayi-Banji, A. and S. Rahman, A review of process parameters influence in solid-state anaerobic digestion: Focus on performance stability thresholds. Renewable and Sustainable Energy Reviews, 2022. 167.

Point 5: Fig:1: Legend is unclear. Resolution should be increased. Choose a different display pattern. Fig. 3 and 4 Legend is unclear. Resolution should be increased.

Response 5: Thanks for reviewer’s kind suggestions. All the figures in manuscript were reinserted with a higher resolution version.

  • On Page 2 of the revised manuscript:

Figure 1. Global primary energy consumption by fuel sources and CO2 emissions

  • On Page 3 of the revised manuscript:

Figure 2. Stages and main pathways in anaerobic digestion

  • On Page 4 of the revised manuscript:

Figure 3. The main pathways of propionate formation in anaerobic digestion (EMP: Embden-Meyerhof-Parnas pathway; LDHA: Lactate dehydrogenase; PCT:Propionyl-CoA transferase; LCD: Lactyl-CoA dehydratase; ACR: acrylyl-CoA reductase; MDH:malate degydrogenase; FH:fumarate hydratase; SDH:succinate dehydrogenase; MCM: methylmalonyl-CoA reductase)

  • On Page 5 of the revised manuscript:

Figure 4. The main degradation pathways of propionate in anaerobic digestion (PTA-ACK: phosphotransacetylase-acetate kinase)

Point 6: Table 1: Scientific names should be in italics.

Response 6: Thank you for your suggestion. Table 1 was modified according to reviewer’s suggestion. And another table in the manuscript was also modified.

  • On Page 5 of the revised manuscript:

Table 1.Degradation reaction of fatty acids and the standard Gibbes free energy

Reaction

△G (kJ/mol, 25 ︒C)

(i) CH3COO-+H2O HCO3-+CH4

-31.0

(ii) CH3CH2COO-+3 H2O CH3COO-+HCO3-+H++3 H2

+76.1

(iii) CH3CH2CH2COO-+2H2O 2CH3COO-+H++2H2

+48.4

(iv) CH3CH2CH2CH2COO-+2H2O CH3COO-+CH3CH2COO-+H++2H2

+25.1

(M) 4 H2+HCO3-+H+ CH4+3 H2O  

-135.6

(ii+M) 4 CH3CH2COO-+3H2O 4 CH3COO-+HCO3-+H++3 CH4

-102.4

  • On Pages 14-15 of the revised manuscript:

Table 2.Summary of reported methaogenic communities using propionate as electron donor

Possible Electrophic microorganism or electron acceptor

Possible Electron donator

Conduit

Employed concentration

References

Methanothrix

Syntrophobacteraceae Thiobacillaceae

Pyrite

5-40 g/L

[137]

Syntrophobacter fumaroxidans

Geobacter sulfurreducens

electric wire

/

[138]

Geobacter, Syntrophobacter, Smithella, and Methanosaeta

Geobacter

coupled effects of ethanol and Fe3O4

500 mg COD/L ethanol + 10 g/L Fe3O4

[139]

Methanothrix

Levilinea

Fe2O3 loaded carbon cloth

/

[140]

Methanospirillum,

Methanosphaerula

Thauera sp.

Magnetite

10-1000 mg/L

[141]

Methanosaeta sp.

Geobacter sp.

biochar

5 g/L

[142]

Methanosaeta sp., Methanosarcina sp.

Geobacter sp.

Electronical conductive pili

[142]

CO2-reducing methanogens

Propionate-oxidazing acetogens

Magnetite

[143]

Methanobacterium

Thauera sp.

Magnetite

20 mM

[61]

Methanosaeta and Methanosarcina sp.

Geobacter

Graphite felt

[144]

Methanobacterium sp.

Anaerolineae and Clostridia

Granular activated carbon

0~5.0g

[145]

Methanosaeta sp.

Propionate and Butyrate

carbon fibers

[146]

Point 7: Instead of piling on the efficacies on AD, authors should discuss about the cumulative carbon conversion efficiency at the end of the process. the manuscript should focus on this aspect only and cite relevant literature.

Response 7: Thanks for your kind suggestion. The data about carbon conversion efficiency was added into the revised manuscript, as well as the relevant literature.

  • On Page 9, Lines 331-333 of the revised manuscript

With the addition of lime mud increased from 2.0 g/L to 10.0 g/L, the carbon conversion rate (carbon in biogas generated from unit feedstock/ carbon in unit feedstock) in-creased by 64.3% (1.4% vs 2.3%).

  • On Page 9, Lines 342-343 of revised manuscript

…with almost no improvement of carbon conversion rate [61].

  • On Page 10, Lines 412-414 of revised manuscript

Supplementation of Fe+Co+Ni to a cow manure anaerobic digester led to an significant increase of carbon conversion rate (from 1.3% to 2.3%) [74].

  • On Page 11, Lines 423-426 of revised manuscript

Assistance of biochar was proved can decrease the dosage of trace elements. Cai et al. [104] found that addition of biochar resulted in a 50 % decrease in trace elements de-mand when the OLR was 5 g TS/L day, with the carbon carbon conversion rate slightly increased from 26.2% to 29.5%.

  • On Page 11, Lines 457-459 of revised manuscript

As result, the carbon conversion rate of food waste anaerobic digestion increased from 3.5% to 3.9% (at OLR of 3.0 g VS/L day).

Point 8: Again, quality of english should be checked by a native speaker. Its not up to the required journal standard.

Response 8: Thanks for the thoughtful suggestion. We have carefully checked and improved the English writing in the revised manuscript.

Round 2

Reviewer 1 Report

Dear authors, you have done good work and I think that this paper can be published in Molecules journal.

Reviewer 2 Report

The manuscript is acceptable in its current format.